# Tumor-Derived Exosomes in Tumor-Induced Immune Suppression

**DOI:** 10.3390/ijms23031461

**Published:** 2022-01-27

**Authors:** Qiongyu Hao, Yong Wu, Yanyuan Wu, Piwen Wang, Jaydutt V. Vadgama

**Affiliations:** 1Division of Cancer Research and Training, Charles R. Drew University of Medicine and Science, Los Angeles, CA 90059, USA; yongwu@cdrewu.edu (Y.W.); yanyuanwu@cdrewu.edu (Y.W.); 2Jonsson Comprehensive Cancer Center, David Geffen School of Medicine, University of California at Los Angeles, Los Angeles, CA 90095, USA

**Keywords:** tumor-derived exosomes, tumor microenvironment, immune suppression

## Abstract

Exosomes are a class of small membrane-bound extracellular vesicles released by almost all cell types and present in all body fluids. Based on the studies of exosome content and their interactions with recipient cells, exosomes are now thought to mediate “targeted” information transfer. Tumor-derived exosomes (TEX) carry a cargo of molecules different from that of normal cell-derived exosomes. TEX functions to mediate distinct biological effects such as receptor discharge and intercellular cross-talk. The immune system defenses, which may initially restrict tumor progression, are progressively blunted by the broad array of TEX molecules that activate suppressive pathways in different immune cells. Herein, we provide a review of the latest research progress on TEX in the context of tumor-mediated immune suppression and discuss the potential as well as challenges of TEX as a target of immunotherapy.

## 1. Introduction

Exosomes are a class of small membrane-bound extracellular vesicles (EVs), typically characterized by their size of 40–150 nm and their expression of marker proteins, including CD63, CD81, and CD9. Exosomes are found to be released by almost all cell types in culture [1,2,3,4,5,6,7] and present in all body fluids [8,9,10,11]. Back in the early 1980s, exosome secretion was thought to be a means to remove cellular waste [12]. Novel roles of exosomes as a critical regulator of cell–cell communications, as well as a potential noninvasive cancer biomarker [13,14,15], have been recently revealed. Based on the studies of exosome content and their interactions with recipient cells, exosomes are now thought to mediate “targeted” information transfer [16].

It is widely accepted that the tumor microenvironment (TME) plays a pivotal role in cancer development and progression [17]. Tumor cells begin to mold the host environment at the early phases of the neoplastic process to favor their proliferation and expansion. Tumor cells were thought to promote this course mainly by pathways involved in cell-to-cell contacts and the release of soluble suppressive factors. However, a novel mechanism has been recently identified involving the active release of immunosuppressive membrane microvesicles, also known as tumor-derived exosomes (TEX). TEX, carrying a cargo of molecules different from that of exosomes made by normal cells, are endosome-derived organelles actively secreted through an exocytosis pathway [18,19]. TEX secretion by tumor cells seems to be a physiological phenomenon that occurs spontaneously. Consequently, TEX functions to mediate distinct biological effects such as receptor discharge and intercellular cross-talk [18,19]. TEX has been linked to a series of functional alterations in the T cells of patients with cancer, ranging from induction of apoptosis to defects in T cell receptor components and functions [20,21,22,23]. In vitro studies showed that TEX were produced by tumor cells in abundance and induced various functional alterations in immune cells [24].

The complexity of networks established between tumor cells and their environment makes it a problematic task to identify potential interventions aimed at disrupting these detrimental connections. This review provides updates to previous studies in this field [25] and discusses the latest research progress on TEX, their cargo, and biological functions in the context of tumor-mediated immune suppression. A timely review of research findings in this rapidly developing field is anticipated to facilitate the decision of future research directions and avoid unnecessary redundancy of work.

## 2. Biogenesis of Exosomes

### 2.1. Secretion and Uptake

Although microvesicles and exosomes have different modes of biogenesis, both entities involve membrane trafficking processes [16]. Briefly, microvesicles originate by an outward budding at the plasma membrane [26]. In contrast, exosomes are generated within the endosomal system as intraluminal vesicles (ILVs) and secreted during the fusion of multivesicular endosomes (MVEs) with the cell surface [27]. For microvesicles, cargoes are enriched in the forming vesicles by a stepwise mechanism of clustering and budding, then followed by fission and vesicle release for secretion within extracellular vesicles. The process of exosome biogenesis begins with the invagination of the plasma membrane to form endosomes. Exosomes are generated as ILVs within the lumen of endosomes during their maturation into MVEs, a process that involves particular sorting machinery. Therefore, exosomes are derived from the endocytic pathway of donor cells [28].

The endosomal sorting complex required for transport (ESCRT) is the most well-established driver of early endosomes (ILVs), which maturate and differentiate into late endosomes within the multivesicular bodies (MVBs) [29]. The presence of ESCRT subunits in exosomes and their machinery in ILV biogenesis opens up a new way of perceiving and understanding the formation of exosomes through manipulation of the ESCRT components (Figure 1). Exosomes can also be generated in an ESCRT-independent manner, which was revealed by studies showing that MVEs, featuring ILVs loaded with CD63, were still formed upon depletion of the four ESCRT complexes [30]. It has been suggested that lysosomes can regulate exosome biogenesis by altering the fate of MVBs [31]. In summary, exosome biogenesis is undoubtedly complex. It seems that both ESCRT-dependent and ESCRT-independent mechanisms operate in exosome biogenesis, and their contributions may vary depending on the cargoes and cell type and can be influenced by other signals and pathological stimuli that the cell can receive.

TEX acquires its cargo from the parent tumor cell via the complex process of biogenesis [32]. ILVs formed in MVBs contain receptors/transmembrane proteins and signal molecules derived from the parent cell surface membrane and the cytosol. The sorting process of these parent cell components is cell specific. TEX, carrying information from the parent tumor cell to recipient cells, is released into the extracellular space when MVBs enclosing pools of future exosomes fuse with the cell membrane [33]. The cargo delivery leads to markedly biological effects, from the cellular transcriptome and proteome to cellular functions in recipient cells [34]. The transmission of exosomes is tissue- and organ-specific in the body. Different integrins expressed on TEX are proved to dictate exosome adhesion to specific cell types and extracellular matrix molecules in particular organs [4]. However, it remains unclear mainly what are the components in the exosomes determine their organ specific location or cell-type specificity. Collectively, the knowledge of vesicle biogenesis, secretion, and uptake is not complete and deserves further exploration.

### 2.2. Morphological and Molecular Features of TEX

#### 2.2.1. Morphology

EVs are heterogeneous in size and functions and comprise a wide variety of poorly characterized vesicular components, including apoptotic bodies (1000–5000 nm), intermediate-sized microvesicles (200–1000 nm), and exosomes (30–150 nm) [35]. Exosomes are different from other EVs because of their distinct biogenesis, which involves the endosomal compartment and is characteristic of all exosomes [36]. Morphologically, exosomes can only be visualized by electron microscopy (EM). TEX is a minor type of EV. TEX resembles other exosomes as spherical, membrane-bound vesicles that measure less than 50 nm in diameter and form aggregates of various sizes [37].

#### 2.2.2. Surface Ligand

Exosomes act as shuttles by transmitting signals and transferring their contents, thus playing an integral role in intercellular communication and regulating physiological and pathological processes of diseases [26,38]. Membrane cargoes are partly derived from the surface of parent tumor cells and endosomes [32], and the sorting of transmembrane shipments into exosomes is mainly dependent on endosomal sorting machinery. The glycosylphosphatidylinositol (GPI)-anchored membrane proteins are present in exosomes, probably because of their affinity for lipid domains and lipid rafts that could be directly involved in ILV generation through their effects on biophysical properties of membranes [39]. Various signaling biomolecules derived from exosome surfaces can functionally regulate multiple cellular processes of recipient cells through interacting with receptor molecules on target cell surfaces [40]. The presence of FasL has been confirmed on the TEX surface [22], which may be surmised that other immune-inhibitory molecules could also be present on the TEX surface.

#### 2.2.3. The Molecular Composition of TEX

Exosomes have emerged as crucial regulators of intercellular communication in cancer. Exosomes released into the TME and body fluids could be taken up by recipient cells through direct fusion of their membrane in different manners such as lipid raft, calveolae, and clathrin-dependent endocytosis, macropinocytosis, and phagocytosis [41,42,43,44]. The intravesicular cargo of exosomes is made up of proteins, lipids, DNAs (mtDNA, ssDNA, dsDNA), and RNAs (mRNA, miRNA, long non-coding RNA, circRNA), which are all functional when transferred into recipient cells [45,46]. Extensive reports of exosome composition have illustrated that exosomes derived from tumors and carrying various cargoes are markedly involved in regulating the biological activities of their recipient cells via the transfer of their oncogenic content that can vary widely between cells and conditions (Figure 2).

##### TEX Protein Content

Proteomic analysis of microvesicles underlined that although several molecules are shared between microvesicles of different cell origins, exosome functionality seems to be determined by specific protein content. It was reported that protein levels of exosome fractions in the plasma of patients with various malignancies correlated with disease activity, tumor grade, tumor stage, response to therapy, and survival [47]. Alterations in levels of TGF-β1 in exosomes isolated from acute myeloid leukemia (AML) patients’ plasma were correlated with patients’ responses to chemotherapy [48].

Extensive proteomic analyses of EVs isolated from cancer patients in the Vesiclepedia databases have shown that TEX’s individual or total protein levels might correlate with cancer development or responses to therapy [49]. These data indicate that the protein signatures of TEX are different from non-malignant cells, and the protein signatures of TEX produced by different tumor cells are also distinct (implying cancer cell-type specificity) [50]. TEX derived from melanoma cells of stage V patients stimulated the formation of a metastatic niche, then encouraged bone marrow-derived cells toward a pro-metastatic phenotype to modulate the metastatic ability of cells via upregulation of the MET oncoprotein [51]. Soluble factors such as cytokines or cytokine receptors could be embedded in the TEX membrane and transported to the recipient cells in *trans* or *cis* configurations, thus expanding and magnifying the immune suppression [52]. Two studies have reported that tumor cells can release TEX enriched in matrix metalloproteinase-13 (MMP-13) and miR-21, thus enhancing metastasis occurring via epithelial-mesenchymal transition (EMT) under hypoxic conditions [53,54].

##### TEX Nucleic Acid Content

Apart from proteins, TEX also carries RNAs, including mRNAs, microRNAs (miRs), and noncoding RNAs [55]. The presence of DNA, mRNA, and miRs in the TEX cargo is essential for the role of TEX as information-carrying vehicles.

mRNA

TEX were reported to contain about 10,000 distinct mRNA species involved in critical cellular activities, including inflammation and immune regulation [56]. TEX isolated from the plasma of patients with recurrent glioma participating in a clinical vaccination trial yielded sufficient quantities of mRNA for quantitative RT-PCR analyses. The mRNA levels of 4 (IL8, TGFB, TIMP1, and ZAP70) of the 24 immune-regulatory genes were significantly decreased in TEX recovered from the paired pre- and post-vaccination plasma samples [57]. Notably, these vaccine-induced changes in the mRNA transcripts occurred only in patients who exhibited immunological and clinical responses to the vaccine. This retrospective vaccination study indicated that measurements of changes in expression levels of immune-related genes in exosomes helped identify vaccine-responsive patients. This study suggests that analysis of mRNA in plasma TEX of cancer patients treated with immune therapies might provide helpful clinical and prognostic information [57].

microRNAs and long non-coding RNA

TEX cargo is rich in miRs, and the miR content of TEX has been extensively investigated [58]. MiRs modulate gene expression in recipient cells by inducing degradation of multiple target mRNAs, depending on the cellular context [59]. The transfer of miRs from tumor cells to immune cells usually downregulates antitumor activities and promotes tumorigenesis [60]. TEX in the plasma of patients with different cancer types carry cancer-specific, distinct miR signatures, which correlate with the cancer development and responses to therapy [18]. Tumor-associated miRs, such as miR-21, miR-146a, miR-155, and miR-568, which have been frequently identified as contents of the TEX cargo, regulate the functions and differentiation of various immune cells [61]. It was determined that TEX derived from sera of breast cancer patients could promote the formation of tumors from nontumorigenic epithelial cells in a Dicer-dependent manner [62]. TEX miR contents also play roles in the induction of normal cell transformation. For example, a study demonstrated that leukemia cell-derived TEX transported miRs (miR-92a) to endothelial cells to modulate endothelial migration and tube formation [63]. Another study reported that prostate cancer cell-derived TEX was involved in tumor expansion through reprogramming of adipose-derived stem cells via oncogenic miRs miR-125b, miR-130b, and miR-155 [64].

In addition to miRs, many species of non-coding RNA are also present in TEX, including vault RNA, Y-RNA, ribosomal RNA (rRNA), and transfer RNA (tRNA) [65,66,67]. Preferential accumulation of specific RNA species appears to occur within TEX [68], suggesting that RNA packaging is not random, but rather mechanisms exist to package specific RNAs into TEX. The RNA processing protein Y-box protein 1 (YB-1) and heterogeneous nuclear ribonucleoprotein A2B1 (hnRNPA2B1) have been implicated in packaging some miRs and non-coding RNAs into TEX through its recognition of RNA sequence motifs [68,69]. Breast cancer cell-derived TEX contain the RNA-induced silencing complex (RISC)-loading complex, including argonaute-2 (Ago2), Dicer, and TAR RNA binding protein (TRBP), associated with miRs [62], which may be an additional mechanism of RNA loading in TEX. It remains unknown if the pathways above are broadly applicable to RNA packaging or if other mechanisms regulating RNA loading exist in TEX [70].

DNA

TEX also contain several types of DNA in addition to RNA species. Mitochondrial DNA (mtDNA) [71,72], single-stranded DNA (ssDNA) [73], and double-stranded DNA (dsDNA) [45,74,75] have been found in TEX. For example, TEX from cancer patients’ plasma and from cultured tumor cells were found to contain double-stranded genomic DNA (gDNA) [76]. TEX can carry and transfer oncogenic mutations to recipient cells [77]. Analyses of gDNA fragments of PTEN, MLH1, or TP53 genes showed that different TEX had distinct gDNA content that could include specific mutations [45,76]. TEX DNA can have functional consequences once transferred into recipient cells transiently [78]. A study showed that TEX DNA was transferred to dendritic cells in a stimulator of interferon genes (STING)-dependent manner [79]. Treatment with topoisomerase-I inhibitors or an epidermal growth factor receptor (EGFR) facilitates DNA packaging into TEX, while the precise mechanisms about DNA packaging in TEX remain to be determined [80].

## 3. TEX-Mediated Immune Suppression

Cancer immunosurveillance is a process of spontaneous cancer immunity and an attempt of the host immune system to restrain cancer growth in the early phases of development [81,82]. However, the equilibrium usually fails with disease progression through escape mechanisms adopted by tumor cells to silence their immunogenic profile and survive by activating immunosuppressive/deviating pathways [83]. Cancer cells are thought to mold microenvironment components and affect immune system function mainly by pathways involved in cell-to-cell contact and the release of soluble suppressive factors, which influence myeloid differentiation [84]. However, the secretion of cytokines and growth factors is not responsible for the totality of the multiple and generalized immune defects in patients with cancer due to rapid degradation by serum proteases in the blood circulation. An alternative novel mechanism is now emerging involving the active release by tumor cells of immune-suppressive microvesicles, such as TEX [85]. TEX offers an efficient vehicle for mediating tumoral immunosuppression. TEX could provide a relatively resistant transporter of bioactive molecules to promote a more effective propagation of tolerogenic signals from the tumor site to distant compartments (Figure 3).

TEX cargo contains elements that induce immune cell dysfunction in different ways to suppress the antitumor immune response [18]. TEX first interacts with immune cells through ligands or antigens, which the cognate receptors on lymphocytes can recognize. TEX directly fuse with the surface membrane, then release their content into the cytoplasm through receptor-mediated uptake. Phagocytic cells such as macrophages and DCs can rapidly take up and internalize TEX. T cells do not seem to internalize TEX readily; instead, TEX interacts with surface molecules to transduce signals that result in sustained Ca^2+^ flux and activation of downstream signaling molecules, leading to alterations in the recipient cell transcriptome [38]. Attempts to link TEX’s molecular and genetic profiles to their immunosuppressive effects, as well as extensive studies of the TEX transcriptome and proteome, are in progress (Table 1). We will highlight a few important studies under each category.

### 3.1. T Cell

A T cell is a type of lymphocyte and plays a central role in the immune response. They originate as precursor cells derived from the bone marrow and develop into several distinct T cells once they have moved to the thymus gland. T cells are grouped into several subtypes based on their function. Groups of differentiated, specific T cells have an essential role in controlling and shaping the immune response by providing various immune-related functions. CD8^+^ and CD4^+^ T cells are selected in the thymus and then undergo further differentiation in the periphery to specialized cells with different functions. CD8^+^ T cells are cytotoxic and can directly kill virus-infected cells as well as cancer cells. Unlike CD8^+^ killer T cells, the CD4^+^ T cells identify and determine if and how the immune system responds to a perceived, specific threat. They indirectly kill foreign cells and function as “helper cells” [159].

It has been reported that TEX-expressing tumor antigens can inhibit T-cell activation and induce apoptosis of T cells. Studies in mice have provided in vivo evidence that the transfer of exosomes from tumor-bearing mice to animals immunized with ovalbumin decreased the activity and frequency of antigen-specific T cells [160]. Whiteside et al. reported that TEX inhibited the proliferation of human CD8^+^ T cells but promoted CD4^+^ T cells ex vivo [18]. Further, TEX induced immune suppression by proapoptosis of antitumor CD8^+^ effector T cells and enhancing suppressor activity of CD4^+^ T regulatory cells, thus contributing to tumor escape [161]. Miyazaki et al. found that TEX from EBAG9-overexpressing prostate cancer cells has the potential to facilitate the immune escape of tumors by inhibiting T-cell cytotoxicity and modulating immune-related gene expression in T cells [162]. In contrast, exosomes derived from normal cells readily induced all T-cell proliferation [163].

#### 3.1.1. Via Surface Ligand

Without being internalized, TEX may deliver their surface ligands to T cell surface receptors to modulate gene expression and functions of T cells [164]. Binding of programmed cell death ligand 1 (PD-L1) to its receptor, programmed cell death protein 1 (PD-1), can lead to the inactivation of cytotoxic T lymphocytes, which is one of the mechanisms for immune escape of tumors [165]. Chen et al. reported that TEX released by metastatic melanomas carry PD-L1 on their surface, which suppressed the function of CD8^+^ T cells and facilitated tumor growth [3]. Poggio et al. observed that exosomal PD-L1 from TEX suppressed T cell activation in the draining lymph node [86]. Ricklefs et al. showed that glioblastoma TEX blocked T-cell activation and proliferation possibly through the binding of PD-L1 expressed on the surface of glioblastoma-derived TEX to the PD1 receptors on T cells [87]. Lero et al. showed that the expression of bioactive FasL and TRAIL on the surface enabled TEX derived from human tumors (such as melanoma and colorectal carcinoma) to induce apoptosis in activated tumor-specific T cells. CD8^+^ T cells are more susceptible to apoptosis by TEX carrying the membrane form of FasL or PD-L1 because of the enrichment of CD95 or PD-1 on the surface of CD8^+^ T cells, respectively [88]. Klibi et al. detected galectin-9 on exosomes in nasopharyngeal carcinoma (NPC) patients’ plasma and NPC mouse xenograft samples. They found that in vivo blocking Tim-3/galectin-9 interaction on exosomes might sustain the anti-tumoral responses of T cells, thereby improving clinical immunotherapeutic efficacy against NPC [89]. In summary, among the various mechanisms responsible for immune suppression, surface ligands of TEX have emerged as significant contributors to tumor growth and tumor escape from the host immune system [166].

#### 3.1.2. Via Other Protein Signals

Other than the surface ligand contact described above, TEX protein cargo can also interfere with T cell functions through the internalization approach. With stochastic optical reconstruction microscopy (STORM) and confocal assay, Wang et al., for the first time, demonstrated that 14-3-3ζ-containing TEX derived from hepatocellular carcinoma (HCC) cells could be swallowed by T cells, resulting in inhibited anti-tumor functions of tumor-infiltrating T cells in HCC microenvironment [90]. Using exosome mass spectrometry analysis, Maybruck et al. revealed that an immunoregulatory protein, galectin-1 (Gal-1), contained in multiple head and neck cancer-derived TEX, was able to induce CD8^+^ T cell suppressor phenotype [91]. In addition, TGF-β included in breast cancer cell-derived TEX was found to be delivered to T cells and decreased T-cell proliferation, which is thought to mediate the hypoxia-induced loss of function of recipient T cells [92]. Czystowska-Kuzmicz et al. reported that TEX, containing a metabolic checkpoint molecule ARG1 found in the ascites and plasma of ovarian cancer patients suppressed the proliferation of CD4^+^ and CD8^+^ T-cells in vitro and in vivo in ovarian cancer mouse models. They found that tumor cells use TEX as vehicles to carry over long distances and deliver ARG1 to immune cells to mitigate the anti-tumor immune responses [93].

#### 3.1.3. Via microRNAs

Ye et al. found that exosomal miR-24-3p was involved in tumor pathogenesis by mediating T-cell suppression via repression of FGF11 and may serve as a potential prognostic biomarker in nasopharyngeal cancer (NPC) Yin et al. observed that TEX derived from mouse sarcoma S-180 cells and Lewis lung carcinoma cells efficiently transported miR-214 to CD4^+^ T cells, resulting in a downregulation of PTEN and Treg expansion [95]. Li et al. showed that oxygen pressure in the TME orchestrated an anti- and pro-tumoral γδ T-cell equilibrium by altering TEX content, which subsequently regulated MDSC function in a miR-21/PTEN/PD-L1-axis-dependent manner [96]. Smallwood et al. demonstrated that autologous patient CD4+ T cells internalized chronic lymphocytic leukemia (CLL)-TEX-containing miR-363 that targets the immunomodulatory molecule CD69. Ye et al. identified five common miRNAs overexpressed in TEX from patient sera or NPC cells: hsa-miR-24-3p, hsa-miR-891a, hsamiR-106a-5p, hsa-miR-20a-5p, and hsa-miR-1908. These over-expressed miRNA clusters down-regulated the MARK1 signaling pathway to alter recipient cell proliferation and differentiation [98]. Bland et al. found that the tumor line B16F0 delivered mRNA/miRNA-loaded TEX to cytotoxic T cells and changed their metabolic function and interferon-gamma production [167]. Together, these results indicate a role of exosomal miRNAs in influencing T-cell functions in TME and may suggest a potential therapeutic modality by integrating exosomal miRNA inhibition and immune checkpoint inhibitor to prevent T-cell dysfunction and enhance the anti-tumor immune responses in cancer treatment.

### 3.2. Natural Killer (NK) Cells

NK cells are innate lymphoid cells involved in protecting the host against infection and cancerous cells and regulating homeostasis via the destruction of activated immune cells [168]. The activity and frequency of NK cells are often suppressed in cancer patients compared with healthy individuals. TEX has been reported to suppress the activity of NK cells to promote the immune escape of cancer cells. Pretreatment of mice with TEX produced by TS/A or 4T.1 murine mammary tumor cells resulted in the accelerated growth of implanted tumor cells in both syngeneic BALB/c mice and nude mice [169]. Mechanistically, the pretreatment with TEX may contribute to the development of tumors by blocking the IL-2-mediated activation of NK cells and their cytotoxic response to tumor cells [169,170].

#### 3.2.1. Via Surface Ligand

Lundholm et al. found that the NKG2D (also known as KLRK1, killer cell lectin-like receptor K1) ligand-expressing prostate tumor-derived TEX selectively downregulated NKG2D on NK and CD8^+^ T cells, leading to impaired cytotoxic function in vitro [99]. Szczepanski et al. found that TEX isolated from AML patients, containing membrane-associated transforming growth factor-β1 (TGF-β1), MICA/MICB and myeloid blast markers (CD34, CD33, and CD117), decreased NK cell cytotoxicity and down-regulated the expression of NKG2D in normal NK cells [48,100]. In contrast, Gastpar et al. found that NK cells pre-incubated with heat shock proteins (Hsp70) surface-positive TEX initiated apoptosis in tumors through granzyme B release [101]. Likewise, Lv et al. showed that HSP-bearing TEX secreted by human hepatocellular carcinoma (HHC) cells under stress conditions efficiently stimulated granzyme B production and NK cell cytotoxicity, along with up-regulated the expression of inhibitory receptor CD94 and down-regulated activating receptors CD69, NKG2D, and NKp44 [102]. In another study, Strandmann et al. identified exosomal nuclear factor HLA-B-associated transcript 3 (BAT3) as a cellular ligand, binding directly and engaging NKp30 on NK cells, triggering NKp30-mediated cytotoxicity in a multiple myeloma model [103].

#### 3.2.2. Via Other Protein Signals

Berchem et al. showed that hypoxia induced a remarkable increase in TGF-β level in TEX derived from K562 (a chronic myelogenous leukemia cell line) and IGR-Heu (a lung carcinoma cell line) cells. The hypoxic TEX transferred TGF-β1 into NK cells, leading to decreased cell surface expression of the activating receptor NKG2D, thereby inhibiting NK cell function [105]. Hong et al. reported that changes in exosomal protein and/or TGF-β1 content might reflect responses to chemotherapy in AML patients [106].

#### 3.2.3. Via microRNAs

Profiling of microRNAs in TEX derived from cancer cells in hypoxic conditions revealed the presence of high levels of miR-210 and miR-23a. Uptake of the hypoxic TEX by NK cells significantly decreased the expression of CD107a, an established marker of NK cell functional activity, in NK cells, which contributed to the impairment of the cytotoxicity of NK cells [105].

### 3.3. Monocytes

Monocytes are a subset of mononuclear leukocytes, which differentiate into macrophages and dendritic cells (DCs) following stimulation by cytokines and other molecules [171]. Monocytes play a significant role in innate and adaptive immunity by producing various effector molecules such as inflammatory cytokines, superoxide, and myeloperoxidase to initiate and contribute toward local and systemic inflammation [172]. Tumor cells and their associated microenvironment can produce molecules such as TEX to alter the recruitment, migration, differentiation, and functional properties of monocytes [173]. Rivoltini et al. reported that co-incubation of peripheral blood monocytes with TEX promoted their differentiation into TGF-β-expressing DCs, which also secreted PGE2 and interfered with cytotoxic T cell generation [85]. Yu et al. demonstrated that TS/A exosomes blocked the differentiation of murine myeloid precursor cells into DCs in vitro [174]. A study on CLL found that CLL-derived TEX played a role in skewing monocytes and macrophages toward a pro-tumorigenic phenotype, which released tumor-supportive cytokines and expressed immunosuppressive molecules such as PD-L1 [175]. Gärtner et al. showed that TEX interacted with primary monocytes and induced an activated phenotype, which was also observed in tumor-associated macrophages [176]. Domenis et al. found that glioma-derived TEX suppressed T cell immune response by acting on monocyte maturation rather than directly interacting with T cells [177].

#### 3.3.1. Via Surface Ligand

Bretz et al. demonstrated that TEX obtained from malignant ascites of ovarian cancer patients significantly induced the secretion of various pro-inflammatory cytokines, such as interleukin (IL)-1β, IL-6, IL-8, and tumor necrosis factor (TNF)-α, via Toll-like receptors 2 (TLR2) and Toll-like receptors 4 (TLR4) binding on monocytes surface, which subsequently activated nuclear factor κB (NF-κB) and STAT3 in the THP-1 human monocytic cells [107]. Fleming et al. showed that TEX from human melanoma cells upregulated PD-L1 expression, leading to immunosuppression of normal monocytes, and the effect was dependent on the surface ligand HSP86 on TEX [108].

#### 3.3.2. Via Other Protein Signals

Song et al. uncovered a mechanism of tumor-associated monocyte survival. They demonstrated that TEX could stimulate the MAPK pathway in monocytes through the transport of functional receptor tyrosine kinase (RTKs), leading to the inactivation of apoptosis-related caspases [109]. Another study showed that TEX could inhibit the differentiation of human monocyte precursors into DCs in colorectal cancer and melanoma. In addition, these monocytes gained the ability to secrete TGFβ, further suppressing T lymphocyte proliferation [178]. Wang et al. showed that GC-derived TEX effectively educated monocytes to differentiate into PD1^+^ tumor-associated macrophages (TAMs) with M2 phenotypic and functional characteristics [179]. Lu et al. provided evidence of a novel mechanism regulating M2 polarization and prostate cancer progression through the transfer of αvβ6 from cancer cells to monocytes through TEX [110]. In contrast, Plebanek et al. have shown that the “non-metastatic” TEX stimulated an innate immune response through the expansion of Ly6C^low^ patrolling monocytes in the bone marrow, which then cause cancer cell clearance at the pre-metastatic niche via the recruitment of NK cells and TRAIL-dependent killing of melanoma cells by macrophages [180].

#### 3.3.3. Via microRNAs and Long Non-Coding RNA

Challagundla et al. identified a role of exosomal miR-21 and miR-155 in the cross-talk between neuroblastoma cells and human monocytes, which contributed to the resistance to chemotherapy through a novel exosomal miR-21/TLR8-NF-κB/exosomic miR-155/TERF1 signaling pathway [111]. Hsieh et al. demonstrated that the EMT transcriptional factor Snail directly activated miR-21 transcription to produce miR-21-abundant TEX, which was engulfed by CD14^+^ human monocytes leading to suppressed expression of M1 markers and increased M2 markers [112]. Likewise, Takano et al. found that TEX carrying miR-203 from CRC cells were incorporated into monocytes and promoted M2 markers’ expression in vitro, suggesting a role of miR-203 in promoting the differentiation of monocytes to M2-TAMs [113]. Van der Vos et al. visualized the release of TEX from glioma cells and their uptake by microglia and monocytes/macrophages in the brain, which resulted in the transfer of miR-451/miR-21 into the recipient cells and supports the functional effects of TEX as a means for the tumor to manipulate its environs [114]. Haderk et al. demonstrated that TEX-mediated transfer of noncoding RNAs to monocytes contributed to cancer-related inflammation and concurrent immune escape via PD-L1 expression in monocytes [175].

### 3.4. Macrophages

Macrophage immune cells have essential roles in antigen presentation, phagocytosis, and immunomodulation. Their functional phenotypes are highly versatile and dependent upon the tissue type and signals presented within its microenvironment, thus allowing macrophages to play multiple roles in the inflammatory process [181]. Activation of M1-phenotype macrophages increases the secretion of pro-inflammatory cytokines and chemokines, leading to immunostimulation and effective elimination of pathogens and infection. In contrast, the M2 macrophages are anti-inflammatory, promote tumor progression, and stimulate angiogenesis and wound healing [182]. The infiltration of TAMs in TME is correlated with tumor development. Various studies have demonstrated that the intercellular communication between cancer cells and TAMs via TEX is able to regulate the phenotype and function of these immune cells.

#### 3.4.1. Via Surface Ligand

Chow et al. revealed that the activation of NF-κB is mediated by the interaction between breast cancer-derived TEX and macrophages, mainly through palmitoylated protein ligands on the surface of TEX and TLR2 on macrophages [115]. Annexin A2, which is highly expressed in breast-cancer-derived TEX and similar to cell surface Anx II, has been reported to promote tPA-dependent angiogenesis, possibly through macrophage-mediated activation of p38MAPK, NF-κB, and STAT3 pathways [116]. Cheng et al. have shown that osteosarcoma cells induced macrophages M2 type differentiation to promote tumor cell EMT through exosomic Tim-3 [117].

#### 3.4.2. Via Other Protein Signals

Gastric tumor-derived TEX was internalized by macrophages and induced an M1 pro-inflammatory response in macrophages through the activation of NF-κB, which stimulated inflammatory cytokines including GCSF, IL-6, IL-8, IL-1β, CCL2, and TNF-α and promoted tumor cell proliferation and migration [118]. With a SILAC-based mass spectrometry strategy, Chen et al. successfully traced the proteome transported from CRC TEX to macrophages. They revealed that the cytoskeleton-centric proteins in CRC TEX played a significant role in transforming macrophages into cancer-favorable phenotypes [119]. Chow et al. found that TEX was internalized by macrophages in axillary lymph nodes, triggering the secretion of pro-inflammatory cytokines such as CCL2, IL-6, TNFα, and GCSF in mice bearing xenograft human breast cancers, and ultimately contributed to metastatic tumor development [115]. Xiao et al. revealed that macrophages were activated after taking up TEX released from oral squamous cell carcinoma (OSCC) cells through p38, Akt, and SAPK/JNK signaling at the early phase. They further found that THBS1 derived from OSCC TEX induced the polarization of macrophages to M1-like TAMs and promoted the migration of OSCC cells [120]. De Vrij et al. investigated the influence of GBM-derived TEX on the phenotype of monocytic cells. Their proteomic profiling showed that GBM TEX was enriched with proteins functioning in extracellular matrix interaction and leukocyte migration. GBM TEX appeared to skew the differentiation of peripheral blood-derived monocytes to alternatively activated M2-type macrophages [121].

#### 3.4.3. Via microRNAs and lncRNAs

The binding of miR-21 and miR-29a from TEX to murine TLR7 and human TLR8 activated NF-κB in macrophages and triggered a TLR-mediated pro-metastatic inflammatory response to promote tumor growth and metastasis [122]. Hypoxic pancreatic cancer cell-derived TEX activated macrophages to the M2 phenotype by delivering miR-301a-3p and activating the PTEN/PI3Kγ signaling pathway in recipient macrophages [123]. Likewise, Chen et al. found that hypoxia induces the expression of miR-940 in TEX derived from epithelial ovarian cancer, which stimulated M2 phenotype polarization to promote cancer cell proliferation and migration [124]. The same group further demonstrated that TEX induced by hypoxia expressed higher miR-21-3p, miR-125b-5p, and miR-181d-5p compared to normoxic TEX, which caused M2 macrophage polarization [125]. MiR-222-3p, enriched in EOC-derived TEX, was also found to increase M2 macrophage polarization and promote angiogenesis and lymphangiogenesis in TME to promote EOC progression [127]. A shift to M2 polarization was also seen in macrophages exposed to TEX released from colon cancer cells harboring gain-of-function mutant p53. These TEX contained high levels of miR-1246, which, when transferred to neighboring macrophages, stimulated the secretion of anti-inflammatory cytokines and EMT-promoting factors and contributed to tumorigenesis and poor prognosis [130]. Xing et al. found that the loss of lncRNA X-inactive-specific transcript (XIST) in breast cancer metastatic brain tumors augmented the secretion of exosomal miRNA-503, which triggered M1-M2 polarization in microglia and contributed to the brain metastasis of breast cancer [131]. Li et al. demonstrated that HCC cell-derived TEX containing elevated levels of lncRNA TUC339 were taken up by THP-1 cells, and TUC339 was subsequently involved in the regulation of macrophage activation [132]. Together, these results suggest the role of TEX miRNAs and lncRNAs in inducing polarization of macrophages to tumor-favorable phenotypes, which in turn promotes tumor proliferation, migration/invasion, and metastasis.

### 3.5. Dendritic Cells

Dendritic cells (DCs) are antigen-presenting cells (APCs) that function to recognize, process, and present antigens on the cell surface to T cells via major histocompatibility complex (MHC) molecules, along with co-stimulatory molecules and cytokines to initiate the immune response [183]. They act as messengers between the innate and the adaptive immune systems. Because of their crucial role in priming specific immune responses, DCs are thought to represent the front line of immune defense that needs to be inactivated to avoid immunity [88]. TEX have been reported to be a potent inhibitor of DC differentiation. Yu et al. demonstrated that the inhibition of DC differentiation in vivo and in vitro was mediated at least partly through TEX-induced IL-6 expression [174].

#### 3.5.1. Via Surface Ligand

Ning et al. found that TEX from a 4T1 breast cancer cell or Lewis lung carcinoma blocked myeloid precursor cells differentiation into CD11c^+^ DCs and induced cell apoptosis. In addition, TEX treatment inhibited the maturation and migration of DCs and promoted the immune suppression of DCs. While blocking, PD-L1 partially restored the immunosuppressive ability of TEX-treated DCs. These data suggest that PD-L1 played a role in TEX-induced DC-associated immune suppression [133]. Dusoswa et al. found that glycan modification of the glioblastoma TEX surface reduced immune inhibitory Siglec binding, while it enhanced TEX internalization by DCs in a DC-specific intercellular adhesion molecule-3-Grabbing non-integrin (DC-SIGN, CD209) dependent manner [134]. Blocking with anti-LFA-1 and anti-DEC205 antibodies or treatment with cytochalasin D could reduce TEX uptake in DCs, suggesting that LFA-1/CD54 and mannose-rich C-type lectin receptor interactions might be critical for the mechanism of TEX uptake by DCs [135]. HSP72 and HSP105 on the TEX surface were found to induce DCs to produce increased IL-6 in a TLR2- and TLR4-dependent manner, which in turn promoted tumor invasion by increasing STAT3-dependent matrix metalloproteinases 9 transcription activity in tumor cells [136]. Porcelli et al. indicated that higher levels of melanoma-derived, uPAR+ EVs in non-responders may represent a new potential target for future therapeutic approaches [184].

#### 3.5.2. Via Other Protein Signals

DCs take up TEXs containing donor antigens, thereby inducing specific CTL responses in vitro or in vivo. Andre et al. found that antigens of TEX could be taken up and cross-presented by MHC-I molecules in HLA-A2+ monocyte-derived DCs [185]. Grange et al. demonstrated that renal cancer cells, particularly cancer stem cells derived TEX impaired the maturation of DCs and T cell immune response by a mechanism involving HLA-G [137]. Salimu et al. identified exosomal prostaglandin E2 (PGE2) as a potential driver of CD73 induction, suggesting a mechanism of DC suppression via exosomal PGE2 [138].

#### 3.5.3. microRNAs/Long Non-Coding RNA/mRNA

Pancreatic cancer-derived TEX were found to transfer miR-203 to DCs, leading to down-regulation of TLR4 expression in DCs and subsequent decrease in TNF-α and IL-12 expression [140]. Asadirad et al. found that TEX was able to deliver miRNA-155 into DCs, which led to an increased expression of surface molecules including MHCII (I/A-I/E), CD86, CD40, and CD83, and increased expression of IL12p70, IFN-γ, and IL10 in DCs, suggesting that miRNA-155 could be a candidate for DC maturation [141].

Chen et al. detected the expression of lncRNAs and mRNAs in DCs treated with pancreatic cancer-derived TEX. They identified 3227 lncRNAs and 924 mRNAs that were differentially expressed, including the LncRNA ENST00000560647 and legumain mRNA, suggesting that TEX may play a critical role in the immune escape of DCs [139].

### 3.6. Myeloid-Derived Suppressor Cells (MDSCs)

MDSCs are a heterogeneous population of immature myeloid cells that mainly consist of precursors of DCs, macrophages, and granulocytes [186]. The differentiation and maturation of these immature myeloid cells are blocked in a pathological environment, especially cancer, which leads to the expansion of MDSCs in vivo [187]. The accumulation of MDSCs during cancer development has emerged as a critical element of cancer-induced immune dysfunction by inhibiting antigen processing and presentation as well as T cell activation, which consequently suppresses immune surveillance and anti-tumor immunity [188]. In TME, TEX released by various tumor cells has recently been demonstrated to play a crucial role in the development, survival, and immunosuppression of MDSCs [189]. Although cargoes conveyed by TEX are various, current studies on functional components of TEXs have revealed that the protein and miRNA contents play a major role in mediating the cell biology of MDSCs [190].

#### 3.6.1. Via Surface Ligand

The interaction of TEX via their membrane HSP ligands with TLR2/MyD88 on MDSCs can activate MDSCs [142]. Diao et al. showed that HSP70 on renal cell carcinoma-derived TEX triggered the activation of STAT3 signaling in MDSCs in a TLR2-MyD88-dependent manner [143]. Likewise, Chalmin et al. showed that HSP72, expressed on a TEX membrane from murine colon carcinoma, mammary carcinoma, and lymphoma, interacted with TLR2/MyD88 on MDSCs and induced immunosuppression of MDSCs by autocrine production of IL-6 through STAT3 [145]. Xiang et al. found that TEX caused IL-6 release from MDSCs in a TLR2/STAT3-dependent manner, whereas TEX re-isolated from syngeneic mice could induce IL-6 in a TLR2-independent way [146]. Gobbo et al. showed that the A8 peptide aptamer could bind to the extracellular domain of TEX membrane HSP70 and block the HSP70/TLR2 association, thereby inhibiting the TEX-induced activation of MDSCs [144].

#### 3.6.2. Via Other Protein Signals

The involvement of TEX proteins in MDSC expansion and immunosuppression has been widely observed. Xiang et al. demonstrated that TEX was taken up by bone marrow-derived myeloid cells, and the resulting cells showed typical phenotypic and functional characteristics of MDSCs. TEX significantly induced the accumulation of MDSCs expressing cyclo-oxygen-ase 2 (Cox2), IL-6, VEGF, and Arg1 and promoted tumor progression via the PGE2 and TGF-β molecules in TEX [147]. In addition to those from solid tumors, TEX from hematological malignancy can also enhance the immunosuppressive capacity of MDSCs. Wang et al. found that, after being taken up by MDSCs, TEX derived from multiple myeloma (MM) cells induced the expansion of MDSCs in vitro and enhanced their accumulation and viability in both murine models and MM patients [149]. Pyzwer et al. demonstrated that using tracking studies, AML-derived TEX was taken up by myeloid progenitor cells, leading to the selective proliferation of MDSCs compared to functionally competent antigen-presenting cells. Mechanistically, the oncoprotein MUC1 induced C-Myc expression and accumulation in TEX, which caused the expansion and proliferation of the target MDSC population through effects on downstream cell cycle proteins [150].

#### 3.6.3. microRNA

TEX enable the direct transfer of nucleic acids involved in cell–cell communication, particularly RNAs [191,192]. Ridder K et al. demonstrated that MDSCs were principal recipient cells for TEX-nucleic acids. Using a Cyclization Recombination Enzyme (Cre)/locus of X-overP1 (LoxP) system to trace exosomal RNAs, they found that MDSCs, after internalizing labeled TEX, displayed enhanced expression of suppressive molecules and altered the miRNA-expressing profile, including the aberrant expression of miR-126-3p, miR-27b, miR-320, and miR-342-3p, which have been reported in the context of tumor progression [151]. Guo X et al. demonstrated that TEX from glioma enhanced suppressive function of MDSCs both in vitro and in vivo, and hypoxia-induced TEX exhibited a more vital ability to induce MDSCs than normoxia-induced TEX. A following mechanistic study revealed hypoxia-induced exosomal miR-29a and miR-92a expression, which in turn activated the expansion and function of MDSCs by targeting HMGB1 and protein kinase cAMP-dependent type I regulatory subunit alpha (Prkar1a), respectively [152]. Ren et al. demonstrated that gastric cancer-secreted TEX delivered miR-107 to the host MDSCs to induce their expansion and activation by targeting DICER1 and phosphatase and tensin homolog (PTEN) genes, suggesting novel therapeutic cancer targets for gastric cancer [153]. Guo et al. found that hypoxia-induced expression of miR-10a and miR-21 in glioma-derived TEX mediated MDSC proliferation and activation through targeting RAR-related orphan receptor alpha (RORA) and PTEN [154]. Li et al. found that oxygen pressure in TME orchestrated an anti- and pro-tumoral γδ T-cell equilibrium by altering TEX content, which subsequently regulated MDSC function in a miR-21/PTEN/PD-L1-axis-dependent manner [96].

In summary, these studies emphasize the importance of TEX in cancer immunosurveillance. Although discrepancies exist, these results suggest that TEX play an essential role in restraining tumor immune surveillance by promoting the immunosuppressive functions of immune cells. The regulatory mechanisms of TEX on cancer immune suppression have also been revealed gradually. These findings are anticipated to boost specific therapeutic targets to eliminate host immunosuppression and enhance the anti-tumor immunotherapy efficacy.

## 4. Future Perspective and Challenges

Strong evidence from in vitro and in vivo animal studies supports the role of TEX in orchestrating an immunosuppressive microenvironment for tumor growth. However, prior to assuming that TEX can be effective targets for immunotherapy, several points need to be considered. First, most preclinical models are unable to simulate the heterogeneity of a tumor, whereas clinically relevant tumors typically contain cancer clones less- or non-responsive to immunotherapy, which evolve to avoid immune-mediated elimination in a process termed “cancer immunoediting”. Second, the relatively short duration of preclinical studies, including animal studies, may not reflect the dynamic process of the immunogenicity of cancer cells that is shaped by the phenotype of the surrounding microenvironment, during which alternative mechanisms of immune evasion may emerge.

Another limitation in assessing the therapeutic potential of TEX thus far is the lack of clinical trial studies. Indeed, the complexity of TEX cargo, as indicated in this review, remains a challenge in developing TEX-targeting immunotherapy. Likely, combination therapies are necessary in order to suppress possible functional compensations among TEX signaling molecules for a long-lasting therapeutic effect. However, increased toxicity may be a concern with combination therapies. Furthermore, as forementioned, in addition to the immunosuppressive microenvironment, the loss of antigenicity/immunogenicity of cancer clones through immunoediting may also contribute to the failure of immunotherapy. This needs to be considered in developing immunotherapeutic strategies based on TEX. In addition, considerable attention has focused on the potential clinical applications of TEVs. However, several technical hindrances have restricted basic and applied research on TEVs. The optimization of the Ti-EV isolation procedures is developing [193].

## 5. Conclusions

Overall, the available results from preclinical analysis of molecular cargo of TEX and their effects on different immune cells support the essential role of TEX in establishing an immunosuppressive microenvironment, which may lead to the modulation of various cancer activities, including invasion, metastasis, angiogenesis, and apoptosis, and induce resistance to immunotherapy. The potential of TEX as therapeutic targets has also been demonstrated in several in vivo animal studies. Despite the limitations/challenges mentioned above, a clear understanding of the molecular profile of TEX and the intricate crosstalks between TEX and immune cells in a tumor microenvironment may lead to effective personalized immunotherapy to improve clinical outcomes. Further research efforts at many levels is needed.

## Figures and Tables

**Figure 1 ijms-23-01461-f001:**
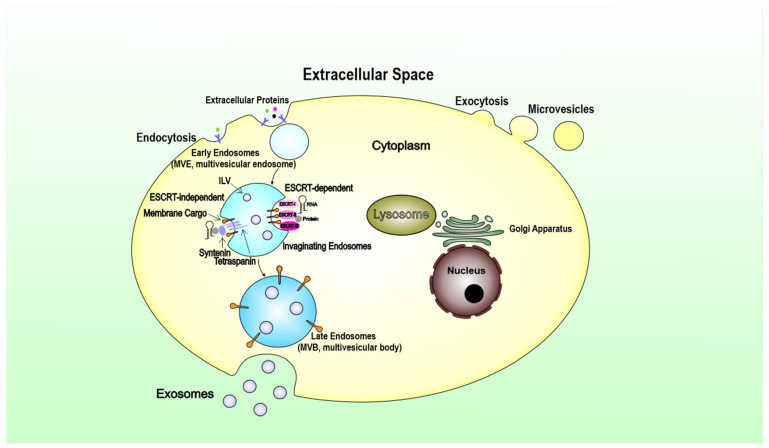
Machinery involved in the biogenesis of extracellular vesicles. Sorting machineries for generating exosomes and microvesicles requires different steps. Membrane-associated proteins and lipids are clustered in discrete membrane microdomains of the multivesicular endosome (MVE) limiting membrane for exosomes. Such microdomains certainly recruit the soluble components, such as extracellular proteins and RNA species by endocytosis. The ESCRT machinery acts in a stepwise manner. ESCRT0 ubiquitylated trans-membrane cargoes on microdomains of MVBs, and ESCRTI subunits cluster, then the soluble components, such as cytosolic proteins and RNA species fating for sorting were recruited via ESCRTII and the ESCRTIII sub-complexes that perform budding and fission. The late endosome MVBs will fuse with the plasma membrane to release the ILVs into the extracellular environment as exosomes by exocytosis.

**Figure 2 ijms-23-01461-f002:**
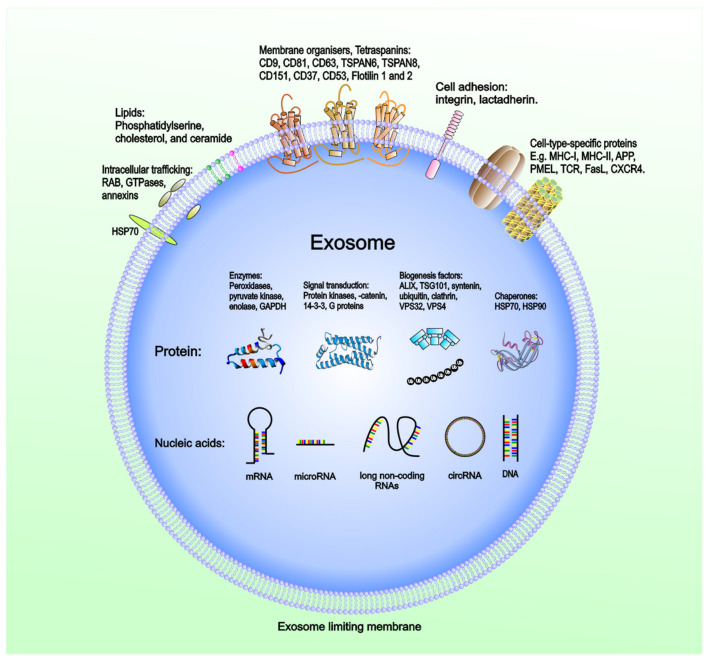
Molecular composition and genetic profiles of TEX. The presence of immune-inhibitory molecules has been confirmed on the TEX surface. The intravesicular molecular composition of TEX is composed of protein (enzymes, signal transducer, biogenesis factors, chaperones, and so on) and nucleic acid (mRNA, miRNA, long non-coding RNA, circRNA, DNA).

**Figure 3 ijms-23-01461-f003:**
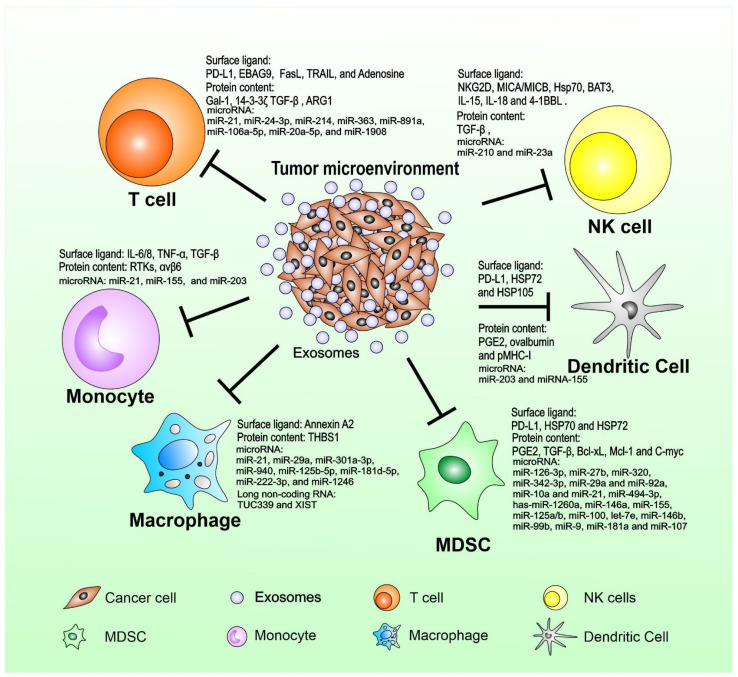
Tumor-released exosomes and their implications in cancer immunity. TEX-mediated signals interfere with immune cell functions at multiple levels and summarize various molecular mechanisms responsible for TEX-mediated effects. The communication network is entirely tumor-driven and designed to promote tumor progression and metastasis by silencing antitumor immune responses.

**Table 1 ijms-23-01461-t001:** TEX’s actions on specific immune cells and underlying mechanisms.

Immune Cell Type	TEX Cargo Component of Effect	Effective Molecules Identified (Reference)
T cell	Surface ligand	PD-L1/PD-1 [3,86,87]; FasL, TRAIL [88]; galectin-9 [89].
	Protein content	14-3-3ζ [90]; galectin-1 [91]; TGF-β [92]; ARG1 [93].
	MicroRNAcontent	miR-24-3p [94]; miR-214 [95]; miR-21 [96]; miR-363 [97]; miR-24-3p, miR-891a, miR-106a-5p, miR-20a-5p, and miR-1908 [98];
Natural killer (NK) cells	Surface ligand	NKG2D [99]; TGF-β1, MICA/MICB, CD34, CD33, and CD117 [48,100]; Hsp70 [101]; HSPs [102]; BAT3 [103]; IL-18, IL-15, and TNFSF9 [104].
	Protein content	TGF-β1 [105,106].
	MicroRNA content	miR-210 and miR-23a [105].
	DNA content	
Monocytes	Surface ligand	TLR2, TLR4 [107]; HSP86 [108].
	Protein content	RTKs [109]; αvβ6 [110];
	MicroRNA content	miR-21 and miR-155 [111,112]; miR-203 [113]; miR-451/miR-21 [114];
Macrophages	Surface ligand	palmitoylated protein ligands [115]; Annexin A2 [116]; Tim-3 [117].
	Protein content	GCSF, IL-6, IL-8, IL-1β, CCL2, and TNF-α[118]; cytoskeleton-centric proteins [119]; THBS1 [120]; proteins functioning in extracellular matrix interaction and leukocyte migration [121].
	MicroRNAs content	miR-21 and miR-29a [122]; miR-301a-3p [123]; miR-940 [124]; miR-21-3p, miR-125b-5p, and miR-181d-5p [125]; miR-let-7b [126]; miR-222-3p [127,128,129]; miR-1246 [130]; miRNA-503 [131];
	Long non-coding RNA content	lncRNA TUC339 [132].
Dendritic cells	Surface ligand	PD-L1 [133]; glycan modification [134]; LFA-1/CD54 [135]; HSP72 and HSP105 [136].
	Protein content	HLA g [137]; PGE2 [138].
	mRNA content	legumain mRNA [139].
	MicroRNA content	miR-203 [140]; miRNA-155 [141];
	Long non-coding RNA content	LncRNA ENST00000560647 [139].
MDSCs	Surface ligand	HSP ligands [142]; Hsp70 [143,144]; Hsp72 [145]; TLR2 [146]; TLR and HSP86 [108]; MyD88 [142];
	Protein content	PGE2 and TGF-β [147]; Mcl-1 and Bcl-xL [148,149]; iNOS [148]; MUC1 [150].
	MicroRNA content	miR-126-3p, miR-27b, miR-320, and miR-342-3p [151]; miR-29a and miR-92a [152]; miR-107 [153]; miR-10a and miR-21 [154]; hsa-miR-494-3p and has-miR-1260a [155]; miR-155 [156]; miR-146a, miR-155, miR-125b, miR-100, let-7e, miR-125a, miR-146b, miR-99b [157]; miR-9 and miR-181a [158].

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
