# Peer review of "Tumor-Derived Exosomes in Tumor-Induced Immune Suppression"

_ijms, 2022, doi:10.3390/ijms23031461_

Round 1

Reviewer 1 Report

The review entitled "Tumor-derived exosomes in tumor-induced immune suppression" by Qiongyu Hao and colleagues addresses a very important question concerning the role of TEX (but also of tumor extracellular vesicles in general) in immune suppression. After a precise and clear description of the biogenesis of exosomes, they deal in detail with the role of TEXs in immune suppression by relating them to every single cell of immunity.

The study is accurate and well organized.

But given the growing acquisitions in the field of exosomes and extracellular vesicles as tumor mediators, my only suggestion is to expand the bibliography with more recent works, such as, for example:

- Tissue-derived extracellular vesicles in cancers and non-cancer diseases: Present and future. Li SR, Man QW, Gao X, Lin H, Wang J, Su FC, Wang HQ, Bu LL, Liu B, Chen G.J Extracell Vesicles. 2021 Dec;10(14):e12175. doi: 10.1002/jev2.12175.

- UPAR + extracellular vesicles: A robust biomarker of resistance to checkpoint inhibitor immunotherapy in metastatic melanoma patients. Porcelli, L.; Guida, M.; De Summa, S.; Di Fonte, R.; De Risi, I.; Garofoli, M.; Caputo, M.; Negri, A.; Strippoli, S.; Serratì, S.; et al. Journal for ImmunoTherapy of Cancer 2021, 9, doi:10.1136/jitc-2021-002372.

- The Importance of Exosomal PD-L1 in Cancer Progression and Its Potential as a Therapeutic Target. Ye L, Zhu Z, Chen X, Zhang H, Huang J, Gu S, Zhao X.Cells. 2021 Nov 19;10(11):3247. doi: 10.3390/cells10113247.

- The roles of exosomal immune checkpoint proteins in tumors. Xing C, Li H, Li RJ, Yin L, Zhang HF, Huang ZN, Cheng Z, Li J, Wang ZH, Peng HL.Mil Med Res. 2021 Nov 8;8(1):56. doi: 10.1186/s40779-021-00350-3.

or other references of the last  year.

In my opinion, this manuscript deserves to be accepted after this minor revision.

Author Response

We really appreciate the reviewer’s comments and suggestions. We have revised the manuscript accordingly, and changes are highlighted in yellow throughout the manuscript. The mentioned references and other references relevant to our topic were added in this revised manuscript.

Reviewer 2 Report

1)  The Nomenclature criteria used to define EVs, Exosomes, and TEX are projected obvious right from the beginning in the text, but what would the terms Exosomes and EV’s and TEX mean to a non-specialist? and do the terms have clear definitions for the specialist? 

2)  The font used in section 3.6 is different, line 103, microdomains of MVBs are written twice.

3)  LINE 575-580; no reference included?  Since the author speaks about TME, cytokines, and miRNA.

Author Response

Response: We really appreciate the reviewer’s comments and suggestions. We have revised the manuscript accordingly, and changes are highlighted in yellow throughout the manuscript.

1)  The Nomenclature criteria used to define EVs, Exosomes, and TEX are projected obvious right from the beginning in the text, but what would the terms Exosomes and EV’s and TEX mean to a non-specialist? and do the terms have clear definitions for the specialist? 

Response: Although confusion on the nomenclature of EVs has spread throughout the literature, EVs may be broadly classified into exosomes, microvesicles (MVs) and apoptotic bodies according to their cellular origin as shown in the table below:

Exosomes

Microvesicles

Apoptotic Bodies

Origin

Endocytic pathway

Plasma membrane

Plasma membrane

Size

40-120 nm

50-1,000 nm

500-2,000 nm

Function

Intercellular communication

Intercellular communication

Facilitate phagocytosis

Markers

Alix, Tsg101, tetraspanins (CD81, CD63, CD9), flotillin

Integrins, selectins, CD40

Annexin V, phosphatidylserine

Contents

Proteins and nucleic acids (mRNA, miRNA and other non-coding RNAs)

Proteins and nucleic acids (mRNA, miRNA and other non-coding RNAs)

Nuclear fractions, cell organelles

TEX, tumor-derived exosomes, carrying a cargo of molecules different from that of exosomes made by normal cells, are endosome-derived organelles actively secreted through an exocytosis pathway.

2)  The font used in section 3.6 is different, line 103, microdomains of MVBs are written twice.

 Response: The font in section 3.6 was corrected. This sentence in line 103 was re-written.

3)  LINE 575-580; no reference included?  Since the author speaks about TME, cytokines, and miRNA.

 Response: The content in this part was re-edited. The references were inserted in the sentences.